# Improving Low-resource Question Answering by Augmenting Question Information

**Andong Chen♠ , Yuan Sun♡, Xiaobing Zhao♡, Rosella P. Galindo Esparza◇**
**Kehai Chen\*† , Yang Xiang♣ , Tiejun Zhao♠ , Min zhang†**
♠ Harbin Institute of Technology, Harbin, China
♡ Minzu University of China, Beijing, China
♣ Pengcheng Laboratory, Shenzhen, China
◇ Brunel University Uxbridge, London, UK
† Harbin Institute of Technology (Shenzhen), Shenzhen, China

## Abstract

In the era of large models, low-resource question-answering tasks lag, emphasizing the importance of data augmentation. The main challenges include leveraging the large model's internal knowledge for data augmentation, determining which QA data component - the question, passage, or answer - benefits most from augmentation and retaining consistency in the augmented content without inducing excessive noise. To tackle these, we introduce PQQ, an innovative approach for question data augmentation consisting of **P**rompt Answer, **Q**uestion Generation, and **Q**uestion Filter. Our experiments reveal that ChatGPT underperforms on the experimental data, yet our PQQ method excels beyond existing augmentation strategies. Further, its universal applicability is validated through successful tests on high-resource QA tasks like SQUAD1.1 and TriviaQA [1].

## 1 Introduction

Low-resource issues pose a significant challenge within the field of Question-Answering (QA) tasks. These problems largely arise from data scarcity and lack of domain-specific training, which often results in underfitting. Even though we are now in the era of large models, which have considerably advanced the field of natural language processing, these low-resource problems in QA tasks persist (Wanjawa et al., 2022; Sun et al., 2021b; Chen et al., 2023a). Consequently, addressing these low-resource issues has become an essential research focus, with data augmentation strategies emerging as a common and effective approach.

However, current research in data augmentation presents several unanswered questions: How can the internal knowledge of large models be harnessed to enhance the quality of augmented data? Which component of QA data - the question, passage, or answer - benefits most from augmentation?

How can we strike a balance between maintaining the consistency of the augmented content and not introducing excessive noise? Despite significant advancements in the field, many of these questions remain unanswered. For instance, The multi-modal QA benchmark, MAQA (Li et al., 2023), accentuates the need for data augmentation using negation examples in QA tasks but doesn't focus on leveraging the knowledge of large models. The fill-in-the-blank QA framework, Gotta (Chen et al., 2023b), leverages generative, prompt-based augmentation methods to enhance learning. However, it does not identify the most beneficial segments of the QA data for improvement. Lastly, the label-guided data augmentation framework PROMPTDA (Chen and Shu, 2022) utilizes label semantics for data augmentation in few-shot learning, however, it overlooks the task of balancing content consistency with noise introduction. Existing augmentation techniques, such as template-based question generation (Ali et al., 2010; Heilman and Smith, 2009; Chali and Hasan, 2015), and the broad application of Seq2Seq models in natural language processing (Bahdanau et al., 2014; Du et al., 2017), have made certain progress but also show similar limitations.

To address these challenges, we propose a novel and straightforward question data augmentation method, termed PQQ. Firstly, we construct prompt templates to generate answers that are related to the original ones but expressed differently (**P**rompt Answer). Then, a diverse set of questions is produced by a question generation model (**Q**uestion Generation). Lastly, using a question filter, we retain only those questions that are logically related to the original QA data as the augmented data (**Q**uestion Filter). Our experimental findings indicate a distinct underperformance of ChatGPT on our experimental dataset, while our proposed methodology, PQQ, exhibits a superior performance, outperforming contemporary mainstream data augmen-

---

\*Corresponding author.

[1]https://github.com/andongBlue/PQQ_QA/

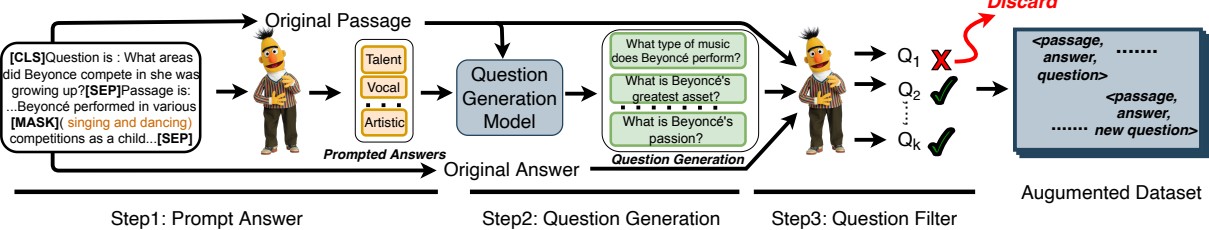

Figure 1: An overview of our approach where the *Bert-large* output Top *k prompted answers* based on original QA data, using *QG* model to obtain *generated questions* by prompted answers and corresponding passage, the *Bert* is employed as a *question filter* to retain questions related to the original dataset.

tation techniques. We further validate the universality of our PQQ method, investigating its efficacy on high-resource question-answering (QA) tasks such as SQUAD1.1 and TriviaQA (Rajpurkar et al., 2016; Joshi et al., 2017). Remarkably, by employing a Bert-base model for QA tasks, our PQQ-augmented approach surpasses the performance of a Bert-Large model without PQQ augmentation, demonstrating the potent efficacy of our methodology.

## 2 PQQ Apporach

This section succinctly outlines the PQQ data augmentation approach. This process, illustrated in Figure 1, consists of three main steps.

### 2.1 Prompt Answer

Implementing prompt-based learning in our research, we transform the QA dataset into a fill-in-the-blank task by substituting answer slots with [MASK] special tokens. Then, we utilize the Bert-Large model, which aligns with our task, to deliver complete phrase outputs. A prompt template, `Input = [CLS]Question is:question[SEP]Passage is:passage`, directs question and passage information to the model. Answer replacements, input starts, and question-passage separators are denoted by [MASK], [CLS], and [SEP], respectively. When inputs exceed the 512-character limit, they're truncated from either end based on [MASK] length.

The Bert-Large model generates various Prompt Answers, with the volume and diversity of these outputs determining the subsequent quantity, diversity, and relevance of questions. For instance, the input *[CLS] Question is: What areas did Beyonce compete in when she was growing up?[SEP] Passage is: ... Beyonce performed in various [MASK] competitions as a child ...* generates four diverse responses(*Talent*, *Artistic*, *Vocal*, and *International*),

enabling the creation of contextually varied questions.

### 2.2 Questions Generation

Once the Prompt Answer is obtained, the next step involves constructing a question generation model to generate a diverse range of question data. Since different QA tasks encompass varying domain knowledge (e.g., TechQA includes education domain knowledge while PolicyQA focuses on legal domain knowledge), it is necessary to fine-tune a specific question generation model for each QA dataset. In our study, we selected the T5-base model (Raffel et al., 2019) as the question generation model.

To fine-tune the question generation model, we utilize passages and answers from the QA dataset as inputs to the T5 model. The format of the input is as follows: *Input = <answer><passage>*. During the fine-tuning process of the T5 model, the loss function employed is a negative log-likelihood estimate.

### 2.3 Question Filter

Figure 1 suggests a possible disconnect between new questions, generated from the Prompt Answer and passage, and the original QA dataset. To tackle this, we repurpose the natural language inference task into a question-filtering task to evaluate the relevance of generated questions to the original data. Using the concept of Bert's pre-trained Next Sentence Prediction task (Devlin et al., 2018; Sun et al., 2021a), we design the Sentence Correlation Prediction (SCP) task to fine-tune the relevance evaluation, which encompasses filter input and output construction.

The SCP fine-tuning task proceeds in two steps. The filter input, **Step 1**, involves the combination of passages and generated questions, following the format $x_{input} =$

$[CLS]Sentence1[SEP]Sentence2$. Here, *Sentence1* is the generated question, *Sentence2* is the sentence containing the answer from the passage, with [CLS] and [SEP] marking the sentence's final hidden state and sentence separation. **Step 2**, filter output, vectorizes the [CLS] output. The SCP header then processes this to calculate the relevance probability as $P_M(n_i \mid Input_i) = \frac{\exp s(n_i|p_i,q_i)}{\sum_n \exp s(n|p_i,q_i)}$. Here, $s = W_{scp}h_{[CLS]}$ is the SCP head, built via an MLP network using $h_{[CLS]}$ vector as input. This results in the class probability distribution, with $p$ and $q$ denoting passages and questions, and $n$ the relevance label. The associated loss function is $L_{NSP} = -\log P_M(n \mid x)$.

| Method | PolicyQA | | TechQA | |
|---|---|---|---|---|
| | EM | F1 | EM | F1 |
| Deep learning methods + Bert-base | | | | |
| Bert-base | 57.3 | 26.9 | 24.6 | 51.7 |
| Bert-Large | **61.1** | 28.7 | 28.8 | 53.9 |
| RoBERTa-base | 58.1 | 27.1 | 24.4 | 48.3 |
| SummAug (2021) | 57.7 | 27.0 | 28.6 | 53.2 |
| LambadaAug(2020) | 57.9 | 26.8 | 24.9 | 49.8 |
| QusAnsAug(2019) | 58.3 | 27.1 | 29.4 | 54.9 |
| AnsAug (2021) | 58.0 | 27.9 | 30.1 | 55.1 |
| MulStaAug (2022) | 58.6 | 28.2 | 31.9 | 58.3 |
| PromptAug(2023b) | 58.2 | 28.8 | 30.8 | 57.9 |
| Large Language Model & Our Approach | | | | |
| ChatGPT(one-shot) | 20.8 | 11.5 | 20.4 | 31.2 |
| Our Approach | 58.9 | **29.7** | **32.6** | **59.8** |

Table 1: The validity experiment results in low-resource datasets. The highest scores of unsupervised methods are in bold.

## 3 Dataset and Experimental Design

In assessing our method's effectiveness on low-resource QA datasets, we chose PolicyQA and TechQA, with 25,107 and 600 training instances respectively, focusing on privacy policies and technology (see Appendix A.1). We conducted two comparative experiments, one applying traditional and SOTA data augmentation techniques in the QA domain, and another employing inference via a large-scale pre-training language model, ChatGPT (details in Appendix A.2). To ascertain the impact of these different approaches, we adopted two evaluation metrics: Exact Match (EM) and F1 scores (Appendix A.3). We trained with Bert-Large and T5 as the prompt answer/question filter and question-generation models respectively. For the ultimate QA tasks, we opted for a fine-tuning approach using a Bert-base model on QA data (training specifics in Appendix A.4). We standardized the augmented data for both datasets to an additional 45,000 <Passage, Answer, Question>

instances and evaluated them using the test set. Appendix **??** provides more examples of the data after augmentation."

## 4 Experiment Result

In this section, we provide a comprehensive analysis of the results to thoroughly evaluate the performance of the proposed approach.

### 4.1 Validity Experiments in Low-resource Datasets

As depicted in Table 1, our data augmentation strategy, referred to as PQQ, has been evaluated on two distinct low-resource Question Answering (QA) datasets: PolicyQA and TechQA. The results highlight the significant advantage of PQQ over comparative methods. Specifically, PQQ achieved an EM score of 58.9 and an F1 score of 29.7 on the PolicyQA dataset, and an EM score of 32.6 and an F1 score of 59.8 on the TechQA dataset, outperforming all comparison methods. The remarkable performance of PQQ can be primarily attributed to its ability to generate high-quality question data.

In terms of data augmentation comparison methods, we have chosen approaches that augment either <Question, Answer> pairs(Schmidt et al., 2022; Alberti et al., 2019) or independent <Answer> data(Van et al., 2021; Chen et al., 2023b). However, even when augmenting the same data, these strategies remain weaker than the PQQ method, thus indicating that augmenting question data exclusively is a better choice.

It is noteworthy that a significant performance gap was observed when testing with ChatGPT[2], a large language model trained on a substantial corpus. Specifically, on the PolicyQA dataset, ChatGPT achieved an EM score of 20.8 and an F1 score of 11.5, while on the TechQA dataset, it scored an EM of 20.4 and an F1 of 31.2. These results emphasize the necessity of fine-tuning low-resource QA data, regardless of the use of broad and diversified training corpora.

| | PolicyQA | | TechQA | |
|---|---|---|---|---|
| | EM | F1 | EM | F1 |
| Bert+PQQ | **58.9** | **29.7** | **32.6** | **59.8** |
| Bert+PQQ-**P** | 58.1 | 29.3 | 31.9 | 59.2 |
| Bert+PQQ-**Q** | 57.8 | 28.4 | 30.9 | 58.1 |
| Bert+PQQ-**P**-**Q** | 57.1 | 27.6 | 29.4 | 57.7 |

Table 2: Results of ablation experiments. **P** indicates the **-Prompt** strategy and **Q** indicate the **-Question Filter**.

---

[2]The ChatGPT used in this work is gpt-3.5-turbo API.

## 4.2 Ablation Experiments

The experimental results, as shown in Table 2, demonstrate the significance of Prompt answers and the Question Filter in the PQQ approach, which was determined through ablation studies conducted by us. Eliminating the Prompt answers template and the Question Filter step resulted in **-Prompt** and **-Question Filter** scenarios, respectively.

Results highlighted that every component of the PQQ framework adds positively to the augmentation results, most notably when employing the full PQQ approach with the Bert-base-based QA model. The Question Filter was particularly influential, as its removal led to a 1 to 2 point reduction in F1 and EM scores. Moreover, the worst results were observed when both **-Prompt** and **-Question Filter** were removed, confirming their crucial roles within the PQQ framework. The experiment results showed a more significant decrease in performance when the **-Question Filter** was removed, demonstrating the effectiveness of using the Question Filter to reduce noise in the augmented data.

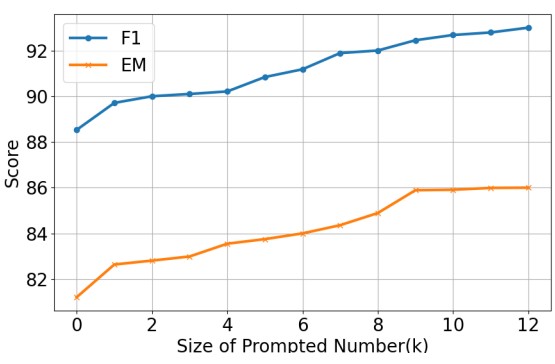

(a) Experimental result on SQuAD1.1

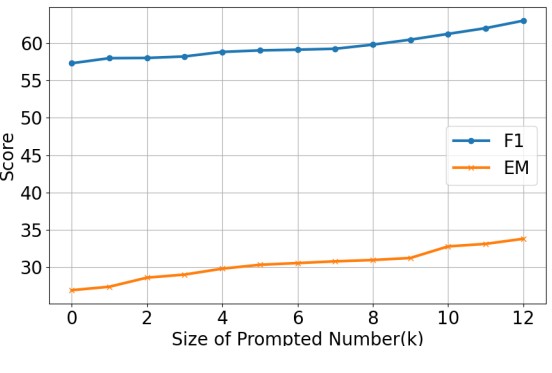

(b) Experimental result on PolicyQA

Figure 2: Experimental results on PolicyQA and SQuAD1.1 using different amounts of prompted answers with a comparison of K-EM and K-F1 scores

## 4.3 Validity Experiments in High-resource Datasets

As shown in Table 3, we applied the PQQ approach to evaluate its effectiveness on both low-resource and high-resource QA datasets, specifically SQuAD1.1(Rajpurkar et al., 2016) and TriviaQA(see Appendix A.1). The our approach technique surpassed baseline measures and demonstrated significant improvement in evaluation metrics. Specifically, when compared to the traditional BiDAF model (Seo et al., 2016), the PQQ approach resulted in an increase of 1.4 points in EM and 1.1 points in F1 for SQuAD1.1. Moreover, training Bert-base with PQQ yielded superior results compared to training Bert-large on the original dataset. Similar enhancements were seen with TriviaQA, where the Bert-base model trained using PQQ outperformed Bert-large trained on the original dataset and surpassed the AugMulStages method (Schmidt et al., 2022) by 2 points in EM and 3.5 points in F1.

| Models | SQUAD1.1 | | TriviaQA | |
|---|---|---|---|---|
| | EM | F1 | EM | F1 |
| No Augmentation | | | | |
| BiDAF | 66.7 | 77.3 | - | - |
| Bert-base | 81.2 | 88.5 | 65.1 | 71.2 |
| Bert-large | 84.2 | 91.1 | 67.9 | 74.8 |
| LLaMA2(one shot) | 19.3 | 28.9 | 18.8 | 31.2 |
| LLaMA2(1.2 epochs) | 27.9 | 38.1 | 27.2 | 55.1 |
| Augmentation | | | | |
| AugMulStages+♠ | 66.9 | 77.6 | - | - |
| our approach+♠ | 68.1 | 78.4 | - | - |
| AugMulStages+♣ | 81.9 | 89.0 | 67.2 | 73.4 |
| our approach+♣ | **86.4** | **92.6** | **69.2** | **76.9** |

Table 3: SQUAD1.1 and TriviaQA experimental results. The Bert-base model and the BiADF(Seo et al., 2016) model are represented by ♣ and ♠ respectively. The version of LLaMA2 is LLaMA2-Chat-7B.The model representation of the subsequent experiments is similar. The highest scores of unsupervised methods are in bold.

## 4.4 Prompt Answer Quantity Analysis

The impact of Prompt Answers quantity on PQQ approach performance, using the F1 metric within SQuAD1.1 and PolicyQA datasets, was investigated. Figure 2 shows that the F1 metric for the high-resource SQuAD1.1 dataset plateaus at 92 with 9 Prompt Answers, suggesting diminishing returns beyond this point. In contrast, the F1 score for the low-sresource PolicyQA dataset escalates past 12 Prompt Answers, implying that more prompts further bolster QA performance in low-resource scenarios. This highlights the crucial influence of Prompt Answers on data augmentation efficiency, with the ideal count differing due to dataset nature

and resource provision. The consistent improvement with more Prompt Answers in low-resource datasets underscores the potential of prompt-based learning in enhancing QA data quality in large pre-trained models.

## 5 Conclusion

Our work presents a new data augmentation method for low-resource question-answering tasks. We first use prompt-based learning to gather various answers. Then, a model generates diverse questions, and a filter retains only those relevant to the original QA data as augmented data. Our approach excels in low-resource contexts and shows broad applicability in the general QA field. Future work will employ large pre-trained models to enhance data quality.

## 6 Limitations

The proposed method is effective for extractive question-answering tasks, but has several limitations. First, its potential in situations where the answers aren't in the provided text remains unexplored. Second, as answer length increases, the employed masking mechanism struggles to cover all words, impacting data quality. This issue awaits further investigation and solution development. Finally, the method's performance hasn't been fully tested on specialized or resource-limited question-answering datasets, necessitating further research to understand its potential in these areas.

## 7 Acknowledgements

We want to thank all the anonymous reviewers for their valuable comments. We also thank Yang Xiang, Xuefeng Bai for their great help and insightful suggestions when polishing this paper. This work was partially supported by the National Natural Science Foundation of China (Grant No.62376075, No.62276077, No.61972436, and No.62106115) and by Shenzhen College Stability Support Plan (Grant GXWD20220811170358002 and GXWD20220817123150002). This work was also partially supported by the International Cooperation Project of PCL (PCL2022D01).

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

## A  Experiment Setup

### A.1  Datasets

To evaluate the effectiveness of our proposed approach on low-resource QA datasets, we selected two representative datasets:

**PolicyQA** (Ahmad et al., 2020): This low-resource question-and-answer dataset focuses on privacy policies. It consists of 25,107 passages derived from the privacy policies of 115 websites, covering various aspects of privacy.

**TechQA** (Castelli et al., 2019): This low-resource QA dataset revolves around the field of technology. The data is collected from real technical forums and comprises 600 training samples, 310 validation samples, and 490 test samples.

**SQuAD1.1**: This is a well-known large QA dataset that contains over 100,000 question-answer pairs from more than 500 Wikipedia articles.

**TriviaQA**: This is a massive and challenging QA dataset with a total of 650,000 QA pairs, surpassing the size of SQuAD 1.1.

### A.2  Comparison Methods

In the realm of natural language processing, we set out to evaluate the effectiveness of our proposed data augmentation approach by comparing it to several existing benchmark strategies. These encompass both deep learning-based techniques such as the augmentation of question data and answer-question pairs.

Methods like SummAug(Lyu et al., 2021) and LambadaAug(Anaby-Tavor et al., 2020) generate fresh questions derived from the original answers. These methods employ language models to ensure the quality of the newly generated data. Alternatively, QusAnsAug(Alberti et al., 2019) and AnsAug(Van et al., 2021) opt to extract and generate novel answers from the original question-answer datasets, providing a different angle of data augmentation.

Another noteworthy approach is Mul-StaAug(Schmidt et al., 2022), which synergizes active learning and data augmentation through the generation of question-answer pairs. The aim of this method is to improve question-answering in low-resource environments. Furthermore, PromptAug(Chen et al., 2023b) utilizes a generative prompt-based method for foundational cloze data augmentation to bolster the learning process. This approach employs a cloze task that involves understanding the context, filling in the blanks,

and subsequently generating answers, simulating the human reasoning process.

In order to ensure a fair evaluation, all of these methods were assessed under identical conditions. We used 45k augmented question-answer instances to level the playing field and negate any bias associated with data volume differences.

### A.3 Evaluation Measures

In the evaluation of the two low-resource QA datasets, quantitative evaluation metrics were employed, namely EM (Exact Match) and F1 scores (Lewis et al., 2019). These metrics were utilized to assess the effectiveness of the data augmentation.

EM (Exact Match): EM calculates whether the predicted outcome exactly matches the standard answer. The EM score is computed using the following formula:

$$EM = \frac{N_{real}}{N_{all}}$$

$N_{real}$ represents the number of predicted answers that exactly match the true answer, while $N_{all}$ represents the total number of true answers.

F1 Score: F1 score measures the word-level match between the predicted outcome and the standard answer. The calculation of the F1 score involves the following steps:

$$P = \frac{N_{overlap}}{N_{allanswer}}$$

$$R = \frac{N_{overlap}}{N_{truthanswer}}$$

$$F1 = \frac{2 \times P \times R}{P + R}$$

$N_{overlap}$ represents the number of words/characters predicted correctly, indicating the lexical overlap between the predicted answer and the true answer. $N_{allanswer}$ denotes the number of words/characters in the predicted answer, while $N_{truthanswer}$ denotes the number of words/characters in the true answer.

### A.4 Training Details

To obtain the Prompt Answer, the first step involves constructing the prompt input template as described in detail in Section 2.1. On the model side, the Bert-large-whole-word-masking (Bert-Large) model provided by Hugging Face[3] is used for inference to obtain the Prompt Answer.

As for the question generation model, the T5 model[4] is initially fine-tuned on the QA dataset for augmentation. During the fine-tuning process, the model is trained using 2 V100 GPUs for 5 epochs, with a learning rate of 7e-4 and a batch size of 8.

For the question filter model, we have opted for the Bert-large-whole-word-masking (Bert-Large) model. Step 2 of fine-tuning the question filter model involves the construction of a three-layered Multilayer Perceptron (MLP) network, functioning as the SCP head, with an intermediate layer dimension of 1024. The Bert-large model is fine-tuned on the QA dataset. As for dataset partitioning, we have allocated 50% of the passage-question pairs from the training set as positive data. The remaining 50% is designated as negative data, accomplished by randomly replacing the question data in the <passage, question> pairs across the entire dataset. The fine-tuning process involves training the model on 1 V100 GPU for 5 epochs, with a learning rate of 5e-4 and a batch size of 4.

The fine-tuning process for the Bert-base model[5] applied to the QA data was conducted using specific parameters. This training regimen entailed a ten-epoch training cycle implemented on a single V100 GPU, with a reduced learning rate set at 1e-5. The batch size was established at eight, and the training was configured to handle a maximum sentence length of 328.

## B supplementary experiments

### B.1 PQQ with LLM

To verify the impact of the current large model on our method, this experiment analyzes the PQQ approach by replacing various components with ChatGPT.For each experimental group, the experiments augmented the training data by 1k on the TechQA dataset. The experimental results are as follows:

The results still prove the effectiveness of our methods. Although ChatGPT has the capability to substitute all PQQ components, its use in a one-shot setting results in heightened data noise and reduced performance. However, employing ChatGPT solely to replace the question filtering component yields a notable boost in performance, indicating its effectiveness for logic consistency classification tasks.

---

[3]https://huggingface.co/bert-large-uncased-whole-word-masking

[4]https://huggingface.co/t5-base

[5]https://huggingface.co/bert-base-uncased

|            | TechQA |      |
|------------|--------|------|
| Technique  | EM     | F1   |
| BERT-base  | 24.6   | 51.7 |
| PQQ        | 26.1   | 53.6 |
| PQQ-replace **P**  | 24.5 | 51.8 |
| PQQ-replace **QG** | 24.9 | 51.7 |
| PQQ-replace **QF** | 25.3 | 52.8 |
| PQQ-replace **ALL**| 24.1 | 51.2 |

Table 4: **P** represents Prompt Answer, **QG** represents Question Generation, and **QF** represents Question Filtering.

## B.2 Augmented Data Volume and Performance Correlation

To further validate the effect of increasing the number of context-question-answer triples, we initiated an additional experiment. Maintaining the same experimental framework as in the Appendix A, the expanded 45,000 samples were divided into batches of 15k, 25k, 35k, 45k, and further extended to 55k, 60k, and 65k. A distinct bert-base model was trained using each subset and subsequently assessed on the SQUAD validation set. The outcomes are presented below:

| Augmented Datas | SQUAD | | PolicyQA | |
|-----------------|-------|------|----------|------|
|                 | EM    | F1   | EM       | F1   |
| 15k | 81.5 | 89.0 | 26.1 | 55.6 |
| 25k | 84.8 | 90.5 | 27.7 | 56.8 |
| 35k | 85.2 | 91.5 | 28.5 | 57.4 |
| 45k | 86.4 | 92.6 | 29.7 | 58.9 |
| 55k | 86.9 | 93.1 | 29.9 | 59.1 |
| 60k | 87.0 | 93.2 | 30.6 | 59.5 |
| 65k | 87.0 | 93.2 | 31.1 | 60.1 |

Table 5: A Comparative Analysis of Augmented Data Volume and its Correlation with Performance.

These results are consistent with the conclusions presented in Figure 2 of our paper. Moreover, this experiment also further validates the effectiveness of our work.

## B.3 Answer Generation Sensitivity to Masking Length

To further investigate this matter, we carried out experiments for validation. Initially, we analyzed the distribution of answer lengths (which equate to mask lengths) in the SQUAD dataset, segmenting them into three groups: [1-3], (3-7], and (7-43], with sample counts of 52,578, 26,086, and 8,935

respectively.

For every length group, we selected 8,935 random QA pairs and employed a consistent PQQ method to produce 10k augmented data points. Distinct bert-base models were trained using each augmented dataset and then assessed on the SQUAD validation set. The outcomes revealed:

|         | SQUAD |       |
|---------|-------|-------|
| Range   | EM    | F1    |
| [1-3]   | 64.31 | 69.96 |
| (3-7]   | 63.98 | 69.31 |
| (7-43]  | 63.04 | 68.47 |

Table 6: Evaluation of Answer Generation Method with Varying Masking Lengths

These findings indeed validate the method's sensitivity. We plan to explore this aspect in greater detail in upcoming research.