# OpenReview forum: "Improving Low-resource Question Answering by Augmenting Question Information"
_EMNLP/2023/Conference — EMNLP 2023 Findings_

### Official Review · Reviewer_KiMC · 2023-08-04

**Soundness:** 3

**Excitement:**

3: Ambivalent: It has merits (e.g., it reports state-of-the-art results, the idea is nice), but there are key weaknesses (e.g., it describes incremental work), and it can significantly benefit from another round of revision. However, I won't object to accepting it if my co-reviewers champion it.

**Paper Topic And Main Contributions:**

To improve Low-resource Question Answering, this paper introduces a approach for question data augmentation consisting of Prompt Answer, Question Generation, and Question Filter.  Firstly, it construct prompt  templates to generate answers that are related to the original ones but expressed differently (Prompt Answer). Then, a diverse set of questions is produced by a question generation model . Lastly, use a question filter to  filter those questions that are logically related to the  original QA data.

**Reasons To Accept:**

Their experiments reveal that ChatGPT underperforms on the experimental data, the proposed PQQ method excels beyond existing augmentation strategies.

**Reasons To Reject:**

This paper is not well-written and lacks some key details

**Reproducibility:**

3: Could reproduce the results with some difficulty. The settings of parameters are underspecified or subjectively determined; the training/evaluation data are not widely available.

**Reviewer Confidence:**

3: Pretty sure, but there's a chance I missed something. Although I have a good feel for this area in general, I did not carefully check the paper's details, e.g., the math, experimental design, or novelty.

---

> ### Author Rebuttal · Authors · 2023-08-29
>
> # Thanks for Reviewer KiMC's valuable comments! Here is the response to Reviewer KiMC.
>
> **Q1**: This paper is not well-written and lacks some key details
>
> **A1**: Thank you for taking the time to review our paper. Regarding the writing issues you've mentioned, I have already sought revisions from native English speakers to improve the quality of the writing in the new version of the paper.
>
> This work focused on introducing a novel data augmentation technique (PQQ) specifically designed for low-resource question-answering tasks. Through prompt-based learning and model-generated queries, our method demonstrates excellent performance in low-resource settings and holds broad applicability across the general QA domain.
>
> In the experiments of the paper, we first explored the effectiveness of PQQ in low-resource (TechQA/PolicyQA) and high-resource (SQUAD/TriviaQA) tasks. Then, through ablation studies, we verified that all components of PQQ have a positive impact. Finally, we examined the influence of the "Prompted answer" hyperparameter on performance in PQQ.
>
> During the review period, we first presented more examples of data augmentation to more clearly display the work's details. We then added comparison experiments with LLMs models and PQQ methods, further proving the effectiveness of our work. Lastly, we conducted a preliminary experiment to analyze the sensitivity of the PQQ method to masking length, thus providing guidance for future research. Additionally, we will release the trained models,  training logs, and code, along with the filtered augmented QA data.
>
> If you have any other concerns, we hope to receive your feedback. We will continue to explain and verify, thereby further improving this work.

---

### Official Review · Reviewer_N3fX · 2023-08-05

**Soundness:** 4

**Excitement:**

4: Strong: This paper deepens the understanding of some phenomenon or lowers the barriers to an existing research direction.

**Paper Topic And Main Contributions:**

This paper proposes an effective framework PQQ for augmenting QA pairs. The proposed method first generates diverse candidate answers by filling masked answer slots and then generates associated questions accordingly. The generated questions are further filtered by an NLI based relevance evaluation module.

**Questions For The Authors:**

A. [CLS] and [SEP] are not the conventional special tokens for T5 models. Are they typos in line 137? If not, have they caused degraded performance?

B. Are the results reported in Tables 1~3 the maximum, mean, etc., or just from a single run?

C. Will you release the code and trained models?

**Reasons To Accept:**

- Comprehensive QA pairs augmentation baselines that are both strong and recent.
- Evaluation was conducted with both high-resource QA datasets (SQUAD1.1 and TriviaQA) as well as low-resource QA datasets (PolicyQA and TechQA)
- The experiment results warrant the effectiveness of PQQ. Table 2 and Figure 2 provide some additional insights for the understanding of the proposed method.

**Reasons To Reject:**

- It remains unclear how the number of available context-question-answer triples used for the answer prompt template would affect the performance of the proposed method.
- The proposed answer generation method is very sensitive to answer/masking length due to the nature of filling masked tokens. (Although the authors list it as future research direction, it could be better if the authors can include some preliminary analysis of this issue in the paper or its appendix)

**Reproducibility:**

4: Could mostly reproduce the results, but there may be some variation because of sample variance or minor variations in their interpretation of the protocol or method.

**Reviewer Confidence:**

4: Quite sure. I tried to check the important points carefully. It's unlikely, though conceivable, that I missed something that should affect my ratings.

**Typos Grammar Style And Presentation Improvements:**

- It could be better if the K-EM and K-F1 shown in figure 2 are plotted with different scales.

---

> ### Author Rebuttal · Authors · 2023-08-29
>
> # Thanks for Reviewer N3fX's valuable comments! Here is the response to Reviewer N3fX.
>
> **Q1**: It remains unclear how the number of available context-question-answer triples used for the answer prompt template would affect the performance of the proposed method.
>
> **A1**: We are delighted to receive this comment. To better understand the work, we initially outline the process of PQQ. Specifically, we generate prompted answers using prompt templates and then use these prompted answers, further refined by a question filter, to augment the question data.
>
> Consequently, line 174 in our paper states that we used the full set of context-question-answer triples for data augmentation, and Section 4.4 discusses the impact of different numbers of prompted answers on performance.
>
> Lastly, to further validate the effect of increasing the number of context-question-answer triples, we conducted another experiment. Keeping the experimental setup consistent with our paper, the augmented 45,000 samples were subdivided into sets of 15k, 25k, 35k, 45k, and further extended to 55k, 60k, and 65k. Each subset was used to train an independent bert-base model, which was then evaluated on the SQUAD validation set. The results are as follows:
>
> |       | SQUAD  |       | PolicyQA |      |
> |-------|--------|-------|----------|------|
> |       |   EM   |  F1  |    EM    |  F1  |
> |  15k  |  81.5  | 89.0 |   26.1   | 55.6 |
> |  25k  |  84.8  | 90.5 |   27.7   | 56.8 |
> |  35k  |  85.2  | 91.5 |   28.5   | 57.4 |
> |  45k  |  86.4  | 92.6 |   29.7   | 58.9 |
> |  55k  |  86.9  | 93.1 |   29.9   | 59.1 |
> |  60k  |  87.0  | 93.2 |   30.6   | 59.5 |
> |  65k  |  87.0  | 93.2 |   31.0   | 60.1 |
> | | | | | |
>
> These results are consistent with the conclusions presented in Figure 2 of our paper. Moreover, this experiment also further validates the effectiveness of our work.
>
> **Q2**: The proposed answer generation method is very sensitive to answer/masking length due to the nature of filling masked tokens. (Although the authors list it as future research direction, it could be better if the authors can include some preliminary analysis of this issue in the paper or its appendix)
>
> **A2**: We are delighted that you are concerned about this phenomenon. To further investigate this matter, we conducted experiments to validate it. We initially examined the distribution of answer lengths (equivalent to the mask lengths) within the SQUAD dataset, categorizing them into three tiers: [1-3], (3-7], and (7-43], with corresponding sample sizes of 52,578, 26,086, 8,935.
>
> For each length category, we randomly sampled 8,935 QA instances and used a fixed PQQ method to generate 10,000 augmented data points. Independent bert-base models were trained on each set of augmented data and were subsequently evaluated on the SQUAD validation set. The results are as follows:
> | Range  |   EM   |   F1   |
> |--------|--------|--------|
> | [1-3]  | 64.31  | 69.96  |
> | (3-7]  | 63.98  | 69.31  |
> | (7-43] | 63.04  | 68.47  |
> | | | |
>
> The experimental results indeed confirmed the sensitivity. In future work, we will further discuss this phenomenon.
>
> **Q3**: [CLS] and [SEP] are not the conventional special tokens for T5 models. Are they typos in line 137? If not, have they caused degraded performance?
>
> **A3**: Thank you for pointing out this detail. In our paper, we used </s> as the delimiter for passages and answers. We will make the appropriate corrections in the new version of the paper. Before settling on this scheme, we did try different approaches and found that various delimiters have some impact on the T5 model's question-generating capability. We fine-tuned the T5 model on SQUAD's training data and used BLEU for evaluation. Previously, we had tried three different separation methods: <1> `w/ </s>`, the structure is `answer</s>passage</s>`; <2> `w/o special tokens`, the format is `answer is: {answer}. passage is {passage}`; <3> `Prompt`, we used `Generate a question related to the answer {answer} within the context of the passage {context}. The experimental results are as follows:
> |                 | BLEU |
> |-----------------|------|
> | w/ </s>          | 21.2 |
> | w/o special tokens | 20.9 |
> | Prompt           | 20.1 |
> | | |
>
> Based on these results, we ultimately chose to use </s> as the delimiter.
>
> **Q4**: Are the results reported in Tables 1~3 the maximum, mean, etc., or just from a single run?
>
> **A4**: In our experiments, we conducted multiple runs to ensure the robustness and credibility of the results. The results reported in Tables 1~3 are the averages obtained from 3 independent runs for each configuration. This approach helps to mitigate the effects of randomness and provides a more robust estimation of the PQQ's performance. We are willing to provide further details in the appendix to facilitate readers and researchers in reproducing our experiments.
>
> **Q5**: Will you release the code and trained models?
>
> **A5**: We will release the trained models, code, and training logs, along with the filtered augmented QA data.
>
> **Q6**: It could be better if the K-EM and K-F1 shown in figure 2 are plotted with different scales.
>
> **A6**: This suggestion greatly aids in enhancing the readability of the paper. We have made the modification in the new version of the figure.

---

### Official Review · Reviewer_xYud · 2023-08-05

**Soundness:** 4

**Excitement:**

3: Ambivalent: It has merits (e.g., it reports state-of-the-art results, the idea is nice), but there are key weaknesses (e.g., it describes incremental work), and it can significantly benefit from another round of revision. However, I won't object to accepting it if my co-reviewers champion it.

**Paper Topic And Main Contributions:**

- This paper aims to augment low-resource qa dataset by using language models.
- The proposed data augmentation method PQQ, consisting of Prompt Answer, Question Generation, and Question Filter, generates additional valid triplets (question, passage, answer) for given data.
- The empirical results show the superiority of PQQ when we train BERT model in a low-resource qa dataset.

**Questions For The Authors:**

- There are experimental results about one-shot Chat-GPT. However, it is possible to replace each fine-tuned model of each step in PQQ to Chat-GPT. In other word, (1) prompt answers using chat-gpt, (2) generate questions using chat-gpt, (3) question filtering using chat-gpt. Can you provide additional experimental results with PQQ (w/ Chat-GPT)?
- Can you provide experimental results when applied to larger language models such as LLaMA-7B?
- If possible, please provide more examples of augmented dataset in the revised version.

**Reasons To Accept:**

- There exist several trials to augment or generate nlp dataset using large language models [1,2]. The proposed QQA suggests that one-shot LLM based dataset generation may underperform when applied to QA data augmentation, showing the value of studying algorithm for efficient and effective QA data augmentation.
- Table 2 is nice.


[1] SQUARE: A Large-Scale Dataset of Sensitive Questions and Acceptable Responses Created Through Human-Machine Collaboration, Lee et al., ACL 2023

[2] WANLI: Worker and AI Collaboration for Natural Language Inference Dataset Creation, Liu et al., EMNLP Findings 2022

**Reasons To Reject:**

- The target model to train is limited.
- I think more ablations are needed. Please check questions.

**Reproducibility:**

3: Could reproduce the results with some difficulty. The settings of parameters are underspecified or subjectively determined; the training/evaluation data are not widely available.

**Reviewer Confidence:**

2: Willing to defend my evaluation, but it is fairly likely that I missed some details, didn't understand some central points, or can't be sure about the novelty of the work.

---

> ### Author Rebuttal · Authors · 2023-08-29
>
> # Thanks for Reviewer xYud's valuable comments! Here is the response to Reviewer xYud.
>
> **Q1**: The target model to train is limited.
>
> **A1**: Based on the situation you've raised, our understanding may not be comprehensive enough due to the lack of context. We currently interpret this situation from two perspectives.
>
> Firstly, we speculate that after PQQ obtains augmented training data, the QA model may suffer from insufficient training. In fact, we train BERT on different tasks until convergence and save the best-performing model on the validation set for final evaluation. Additionally, we will release the source code, model weights, training logs, and augmented QA data available for researchers to refer to.
>
> Secondly, it is considered that the QA task in this work is limited to extractive models (like BERT) rather than generative models (such as LLaMA), thus imposing limitations. Regarding this perspective, we conducted two supplementary experiments where we replaced various components of PQQ with ChatGPT and other LLM models (LLaMA2-Chat-7) on QA data. The results of both experiments have proven that current extractive models have advantages in the QA tasks mentioned in the paper.
>
> If there are aspects of your comment that we may not have fully grasped, please let us know. We look forward to your feedback to help improve this work.
>
> **Q2**: There are experimental results about one-shot Chat-GPT. However, it is possible to replace each fine-tuned model of each step in PQQ to Chat-GPT. In other word, (1) prompt answers using chat-gpt, (2) generate questions using chat-gpt, (3) question filtering using chat-gpt. Can you provide additional experimental results with PQQ (w/ Chat-GPT)?
>
> **A2**: This is a very interesting idea, and we have conducted experiments according to your recommendation. For each scheme, we augmented 1k QA data (TechQA comprises 600 training and 490 samples) and separately trained BERT models for validation on the test set, with the results as follows:
> | Technique      | EM  | F1  |
> |----------------|-----|-----|
> | BERT-base      | 24.6| 51.7|
> | PQQ            | 26.1| 53.6|
> | PQQ-replace *P*  | 24.5| 51.8|
> | PQQ-replace *QG* | 24.9| 51.7|
> | PQQ-replace *QF* | 25.3| 52.8|
> | PQQ-replace ALL| 24.1| 51.2|
> Note: *P* represents Prompt Answer, *QG* represents Question Generation, and *QF* represents Question Filtering.
>
> Excitingly, the results still prove the effectiveness of our methods. Additionally, we gained some insights. First, although ChatGPT can replace all PQQ components, using it in a one-shot setting leads to increased data noise and decreased performance. Secondly, when ChatGPT is only used to replace the question filtering component, there is a significant improvement in performance, suggesting that it is useful for logic consistency classification tasks.
>
> **Q3**: Can you provide experimental results when applied to larger language models such as LLaMA-7B?
>
> **A3**: Thank you for your suggestion. We have successfully applied it to low-resource (TechQA) and high-resource (SQUAD) tasks in LLaMA2-Chat-7B. The experimental results are as follows:
>
> |                   | EM   | F1   |
> |-------------------|------|------|
> | SQUAD (one-shot)   | 19.34| 28.90|
> | SQUAD (1.2 epochs) | 27.89| 38.12|
> | TechQA (one-shot)  | 18.78| 31.17|
> | TechQA (1.2 epochs)| 27.19| 55.10|
> | | | |
>
> These results align with those presented in Table 1 of our paper. The findings underscore the importance of fine-tuning on these QA datasets. The complete set of results will be furnished in the final version of the paper.
>
> **Q4**: If possible, please provide more examples of augmented dataset in the revised version. seems to be quite important.
>
> **A4**: We appreciate your feedback. To better understand augmented data, here are some additional examples:
> | Passage | Original answer | Original question | Prompted answer | Generated question |
> |---------|----------------|-------------------|-----------------|--------------------|
> | [...]It is a replica of the grotto at Lourdes, France where the Virgin Mary reputedly appeared to Saint Bernadette Soubirous in 1858. [...] | { "text": [ "Saint Bernadette Soubirous" ], "answer_start": [ 515 ] } | To whom did the Virgin Mary allegedly appear in 1858 in Lourdes France? | Bernadette | To whom did the Virgin Mary reputedly appear in the grotto at Lourdes, France in 1858, according to the description of the architectural features of the school? |
> | [...]It is a replica of the grotto at Lourdes, France where the Virgin Mary reputedly appeared to Saint Bernadette Soubirous in 1858. [...] | { "text": [ "Saint Bernadette Soubirous" ], "answer_start": [ 515 ] } | To whom did the Virgin Mary allegedly appear in 1858 in Lourdes France? | Soubirous        | What is the last name of Saint Bernadette, to whom the Virgin Mary reputedly appeared in Lourdes, France in 1858? |
> | [...]It is a replica of the grotto at Lourdes, France where the Virgin Mary reputedly appeared to Saint Bernadette Soubirous in 1858. [...] | { "text": [ "Saint Bernadette Soubirous" ], "answer_start": [ 515 ] } | To whom did the Virgin Mary allegedly appear in 1858 in Lourdes France? | Someone          | To whom did the Virgin Mary reputedly appear at the grotto in Lourdes, France, according to the description of the school's architectural features? |
> | | |  |
>
> From the examples, you can see that the prompted answer is related to the original answer, and the generated questions are logically consistent with the original QA data. Due to space limitations, we will list more examples in the final appendix.

---

### Meta-Review · Area_Chair_6MQg · 2023-09-15

**Recommendation:** 3

**Metareview:**

The paper tackles to improve QA performance by generating synthetic question-answer pairs and training on them. This method is demonstrated to improve two low-resource QA datasets as well as two high-resource QA datasets.

Reviewers pointed out that experiments support the effectiveness of the method well (xYud, N3fX, KiMC), baselines are comprehensive (N3fX) and analysis is insightful (xYud, N3fX).

Reviewers raised concerns on the sensitivity of the method (N3fX) and the presentation (KiMC). Concerns on lack of ablations were also raised (xYud) which are addressed in the author responses.

During the reviewer discussion period, reviewers raised additional concerns on limited novelty, given that the general pipeline of answer identification -> question generation -> filtering for data augmentation in QA has been done in prior work [1, 2, 3]. However, the paper still made a small-but-non-trivial contribution on improving answer identification which goes beyond simply using linguistic rules and encourages the diversity of the answers, which is well-supported by a set of experiments and additional ablations provided during the rebuttal period.

[1] https://arxiv.org/pdf/2004.11546.pdf
[2] https://arxiv.org/pdf/2010.12643.pdf
[3] https://arxiv.org/pdf/2102.07033.pdf

---

### Decision · Program_Chairs · 2023-10-07

**Decision:**

Accept-Findings

**Comment:**

The paper tackles to improve QA performance by generating synthetic question-answer pairs and training on them. This method is demonstrated to improve two low-resource QA datasets as well as two high-resource QA datasets.

Reviewers pointed out that experiments support the effectiveness of the method well (xYud, N3fX, KiMC), baselines are comprehensive (N3fX) and analysis is insightful (xYud, N3fX).

Reviewers raised concerns on the sensitivity of the method (N3fX) and the presentation (KiMC). Concerns on lack of ablations were also raised (xYud) which are addressed in the author responses.

During the reviewer discussion period, reviewers raised additional concerns on limited novelty, given that the general pipeline of answer identification -> question generation -> filtering for data augmentation in QA has been done in prior work [1, 2, 3]. However, the paper still made a small-but-non-trivial contribution on improving answer identification which goes beyond simply using linguistic rules and encourages the diversity of the answers, which is well-supported by a set of experiments and additional ablations provided during the rebuttal period.

[1] https://arxiv.org/pdf/2004.11546.pdf
[2] https://arxiv.org/pdf/2010.12643.pdf
[3] https://arxiv.org/pdf/2102.07033.pdf